# Cytotoxicity of Biodegradable Zinc and Its Alloys: A Systematic Review

**DOI:** 10.3390/jfb14040206

**Published:** 2023-04-07

**Authors:** Qian Liu, An Li, Shizhen Liu, Qingyun Fu, Yichen Xu, Jingtao Dai, Ping Li, Shulan Xu

**Affiliations:** 1Center of Oral Implantology, Stomatological Hospital, School of Stomatology, Southern Medical University, Guangzhou 510280, China; 2The School of Computing Science, University of Glasgow, Glasgow G12 8RZ, UK; 3State Key Laboratory of Oral Diseases & National Clinical Research Center for Oral Diseases, Department of Oral Prosthodontics, West China Hospital of Stomatology, Sichuan University, Chengdu 610041, China

**Keywords:** zinc, zinc alloy, biodegradable metals, absorbable metals, cytotoxicity

## Abstract

Zinc-based biodegradable metals (BMs) have been developed for biomedical implant materials. However, the cytotoxicity of Zn and its alloys has caused controversy. This work aims to investigate whether Zn and its alloys possess cytotoxic effects and the corresponding influence factors. According to the guidelines of the PRISMA statement, an electronic combined hand search was conducted to retrieve articles published in PubMed, Web of Science, and Scopus (2013.1–2023.2) following the PICOS strategy. Eighty-six eligible articles were included. The quality of the included toxicity studies was assessed utilizing the ToxRTool. Among the included articles, extract tests were performed in 83 studies, and direct contact tests were conducted in 18 studies. According to the results of this review, the cytotoxicity of Zn-based BMs is mainly determined by three factors, namely, Zn-based materials, tested cells, and test system. Notably, Zn and its alloys did not exhibit cytotoxic effects under certain test conditions, but significant heterogeneity existed in the implementation of the cytotoxicity evaluation. Furthermore, there is currently a relatively lower quality of current cytotoxicity evaluation in Zn-based BMs owing to the adoption of nonuniform standards. Establishing a standardized in vitro toxicity assessment system for Zn-based BMs is required for future investigations.

## 1. Introduction

Increasing attention has been dedicated to biodegradable metals (BMs) due to their potential to replace permanent implant materials for those of temporary function. BMs, mainly those based on magnesium (Mg), iron (Fe), zinc (Zn), and their alloys or composites, are expected to degrade gradually and leave no residues in vivo [1,2,3]. Zn-based BMs have been developed and investigated as potential implant materials since they have a moderate degradation rate and superior mechanical properties, which are suitable for clinical applications. Bowen et al. conducted a landmark study on the in vivo performance of pure Zn, demonstrating its excellent in vivo biocompatibility and suitable biodegradability for cardiac stent applications [4]. These findings have inspired researchers from various fields to investigate Zn-based BMs.

Currently, the main focus of the clinical application of novel Zn-based BMs lies in their use as vascular stents [5,6], surgical sutures [7,8], and craniomaxillofacial and orthopedic implants [9]. Excellent biocompatibility is a prerequisite for biomedical implant materials. Unlike Ti and its alloys, whose biocompatibility has been established for medical use [10,11], the toxicity of Zn and its alloys remains a subject of debate. Considering the 3Rs principles in animal research, i.e., to reduce, refine, or replace the use of animals in biomedical research, the in vitro assessment is indispensable in estimating the biocompatibility of the novel Zn-based BMs [12,13]. Although ISO 10993-5/12 standards provided a rapid and sensitive approach to assess the potential toxicity of substances [14], conflicting reports on the cytotoxicity of Zn-based BMs have thrown their biosafety into further confusion [15,16].

This systematic review aims to clarify the cytotoxicity of Zn and its alloys according to evidence-based biomaterials research retrieved through an extensive and comprehensive search strategy [17].

## 2. Materials and Methods

### 2.1. Quality Assurance and Criteria

This review was conducted in accordance with the Cochrane Handbook for Systematic Reviews of Interventions and is based on the handbook from the Office of Health Assessment and Translation (OHAT—NIH) for in vitro toxicological studies. We searched for the study articles according to the PICOS (patient, intervention, comparison, outcome, and study design) framework and reported according to PRISMA guidelines (Appendix A). Two researchers strictly followed the PICOS strategy and extracted data independently after reviewing the titles, abstracts, and full-text articles. The quality assessment was performed using the Toxicological Data Reliability Assessment Tool (ToxRTool).

### 2.2. Search Strategy

The PICOS framework was followed as the basis of a search strategy, involving the following factors:Population (P): cells.Intervention (I): biodegradable Zn and its alloys.Comparison (C): nonbiodegradable metals, such as stainless steel, titanium, titanium alloy, and cobalt–chromium alloy; biodegradable polymers, such as polylactic acid; other biodegradable metals, such as Mg-based BMs.Outcome (O): cell viability.Study design (S): in vitro study.

An electronic search was performed utilizing the PubMed, Scopus, and Web of Science databases of articles published up to 28 September 2022. The search was implemented using a combination of medical subject heading (MeSH) terms and free words. The search strategies that were developed for each database are given in Appendix A. Furthermore, a hand search of reference lists for potential eligibility of included articles was performed. A supplementary search was conducted on 1 February 2023 to update the references in a timely fashion.

### 2.3. Inclusion and Exclusion Criteria

The inclusion criteria were studies in the English language that conducted cytotoxicity assessment of biodegradable Zn or its alloys, in which experiments were implemented according to the ISO (International Organization for Standardization) 10993-5 or 10993-12 standards. The exclusion criteria were review articles, clinical studies, in vivo animal studies, case reports, retrospective studies, editorials, opinions, guidelines, conferences, and commentary articles.

### 2.4. Study Selection and Data Extraction 

The database search outputs, generated using established search strategies, were imported into Endnote (Version X9.1) to remove the duplicate publications, and then the Rayyan website was used for blinded screening. Two researchers (Q.L. and P.L.) independently screened the literature and extracted the data in strict accordance with the inclusion/exclusion criteria, and then the preestablished data extraction checklist. Disagreements on the eligibility of studies were resolved through careful discussion, and any remaining disputes were resolved by a third researcher (A.L.).

For data extraction, the pre-established data extraction table, which contained basic and experimental information, was utilized for data extraction and analyzed by Microsoft Office Excel 2013. The basic information comprised the year of publication and the author(s). The extracted experimental data were divided into three parts: the Zn-based material, tested cells, and test system. These items included test materials, material processing, types of cell lines, test methodologies, extract ratio and immersion time, dilution ratio, exposure time, use of controls, selected assays, other parameters, and study outcomes.

### 2.5. Assessment of Quality of Evidence

Two researchers (Q.L. and P.L.) independently conducted the reliability assessment using the ToxRTool, which was developed by the European Center for the Validation of Alternative Methods (ECVAM). ToxRTool provides comprehensive criteria and guidance for evaluating the inherent quality of toxicological data or reliability at the methodological level. The in vitro part of this tool consists of an 18-point rating checklist, which is grouped into the following five aspects: (1) test substance identification; (2) test substance characterization; (3) study design description; (4) study results documentation; (5) plausibility of study design and data. Each criterion can be graded as “1” (i.e., “criterion met”) or “0” (i.e., “criterion not met” or not reported). Then, the results of 18 criteria were combined to determine the overall quality of the included articles. According to the reliability categorization, articles with 15–18 points were considered reliable without restrictions, studies with 11–14 points were reliable with possible restrictions, and studies with fewer than 11 points were considered unreliable. Furthermore, according to ToxRTool, items 1, 8–12, and 17 were highlighted in red, indicating particular importance. Regardless of the quality assessment score, studies that failed to meet all of the abovementioned red item criteria were classified as unreliable.

## 3. Results

### 3.1. Included Studies

The selection process for the included articles is presented in Figure 1. According to the inclusion and exclusion criteria, a total of 86 articles published between 2013 to 2023 were included in this review. The main characteristics of the included articles are summarized in Table 1.

### 3.2. Quality Assessment According to the ToxRTool

Detailed total scores of each ToxRTool item in vitro criteria are presented in Figure 2a. Twenty-nine studies described the source of test substances. A total of 34 and 35 studies were graded as “1” in items 12 and 13, respectively. A total of 40 studies provided the source information of the test system. In addition, 55 studies set negative controls in cytotoxicity evaluation, and 73 articles met the 16th criterion detailing the statistical method for data analysis. The number of articles that met the remaining criteria ranged from 80 to 86 (over 90% of the total articles). As depicted in Figure 2b, it is clear that the quality assessment of included toxicity data was found to be relatively low. Furthermore, 31 articles were reliable without restrictions, three were deemed reliable with restrictions, and 52 articles were not reliable. All studies classified as not reliable failed to meet all of the essential criteria marked in red in the assessment tool. 

### 3.3. Main Characteristics of the Included Articles

#### 3.3.1. Materials and Processing

As shown in Table 1, 51 articles involved pure Zn, and 77 investigated Zn-based alloys. The latter mainly consisted of binary and ternary alloys such as Zn–Mg [15,25,26,27,28,35,43,53,57,70,82], Zn–Cu [23,24,29,37,44,56,60,71,75,80,81,86], Zn–Ag [36,42,66,69], Zn–Mn [41,51,90], and Zn–Li alloys [38,61,64], with Zn–Mg and Zn–Cu being the predominant types. A small portion of Zn alloys comprised additional Al [47], Fe [49,50,84,85], Ca [58,77,88], Ge [55], Ti [54,87], Sr [65,72,76], Si [84], Zr [31,33], Sn [47], and V [59]. In addition, pure Zn was alloyed with rare-earth elements (REEs) such as Er, Dy, Ho, Ce, and Gd [57,68,78].

The morphological geometry of the test samples was varied, including discs, porous scaffolds [37,39,53], various thread types [60,70], foil [60], wires [40], membranes [85,87], and cylinders [25]. In terms of the preparation and processing of test samples, casting, hot-rolling, and hot-extruding were the main manufacturing processes for biodegradable Zn and its alloys. Other processing technologies included sintering [15], electrodeposited heat treatment [37], and additive manufacturing [39,47,53,68].

#### 3.3.2. Tested Cell Types

Figure 3a displays the division of cell types tested into three main categories: cells pertaining to osteoblasts, cardiovascular-related cells, and others. A large proportion of studies used immortalized cell lines in evaluating toxicity, whereas only five used primary cells (with passage number ≤7) [21,22,34,46,69]. Regarding orthopedic investigations, 21 selected articles used the MC3T3 (pre-osteoblast cell line), which is usually used in osteogenic evaluation. Although the vast majority used MC3T3-E1, one report used another clone number [69] and one did not describe the specific clone number [66]. Fourteen studies adopted MG-63 in cytotoxicity evaluation, a cell line from human osteosarcoma, and nine studies investigated Saos-2 osteoblasts (human primary osteosarcoma cell line). A few studies used other cell types, such as bone marrow mesenchymal stem cells (MSCs) [30,46,67,91]. Human umbilical vein endothelial cells (HUVECs) and human endothelium-derived cell lines (EA.hy926) were the most commonly employed cell lines in vascular research. The mouse fibroblast cell line (L929) was investigated in 24 articles, accounting for a considerable fraction of the third category. Overall, more than 70% of the articles evaluated only one cell line, 19 articles (22.9%) compared the toxicity performance of two cell lines, and five studies (6.0%) selected three different cell lines in carrying out cytotoxicity tests.

#### 3.3.3. Test System

Concerning the extraction parameter, for evaluating the cytotoxicity of Zn-based BMs, 68 publications employed extract tests, while three studies exclusively conducted direct contact tests [33,74,77]. The remaining 15 studies combined the two approaches (Figure 3b). As shown in Figure 3c, various sample surface-to-volume extraction ratios and immersion times were used to prepare the sample extracts. Nineteen extract tests applied the ratio of 1.25 cm^2^/mL with sample immersion times of 72 h and 24 h, respectively. Eight studies were conducted using the sample powder to evaluate cytotoxicity and adopted the extraction ratio of sample weight to medium volume [19,22,37,39,41,53,61,70]. Notably, 13 studies did not specify a certain ratio. Additionally, some researchers adopted the extraction ratios such as 3 mL/cm^2^, 1.25 mL/cm^2^, and 20 mL/cm^2^ without providing any justification [32,52]. Extracts were often filtered via a membrane [19,20,39,49,51,53,60,65,69,72] or centrifuged to withdraw the supernatant fluid in certain studies [18,20,26,40,58,62,80,101,102]. Some studies investigated Zn-based samples precultured in the medium before the cytotoxicity test [25,28,72,74,77,92,102]. The effect of BSA (bovine serum albumin) on the cytotoxicity in the medium of pre-exposure samples was also examined [72]. Jablonska et al. confirmed the effect of FBS (fetal bovine serum) in the cell medium on cytotoxicity tests [15,83]. The effect of pretreatments such as pre-cultivation, stabilization treatment [91], sterilization treatment [75], sandblasting [93], and acid etching treatment [102] was also investigated. 

Regarding the concentration of the extracts, approximately 76% of studies (63/83) set concentration gradients by dilution with the cell medium, while 18.4% (14/83) used undiluted extracts solutions for testing. It was ambiguous whether the extracts were diluted in six articles [68,70,77,81,87,90].

As shown in Figure 3d, cell viability tests could be divided into qualitative and quantitative assays. Among the assays used for quantitative tests, tetrazolium salt-based assays such as CCK-8 (WTS-8), MTT, WTS-1, and MTS were used frequently. Forty-seven studies used CCK-8 assays in cytotoxicity evaluation, ranking first. Cell survival was determined using the MTT assay in 19 studies and the MTS assay in eight studies. Two studies were tested using the lactate dehydrogenase (LDH) release assay, while five were tested using the bromodeoxyuridine (BrdU) incorporation assay to measure cell proliferation. In a few studies, cell viability was determined on the basis of a fluorometric resazurin reduction method, such as CyQUANT [46] and resazurin assays [15,33]. The assays for qualitative analyses are summarized in Figure 3e, in which live/dead staining FDA/EB dye [16,44,59], Calcein-AM/PI dye [30,82,101], and FITC–phalloidin/DAPI dye [25,51,58,64,65] were the primary assays to realize the visualization of cells. Six studies did not indicate the specific assays used in qualitative assessment.

With regard to the control groups, 32 studies set both positive and negative control groups, while 28 studies set only negative control groups, and the remaining 26 studies set neither negative nor positive control groups. The cells in the cell culture medium alone and supplemented with 5–20% dimethyl sulfoxide (DMSO) were usually chosen as negative and positive control groups, respectively. Some researchers also used Cu and Ti–6Al–4V alloy as negative and positive controls, respectively [59,73,74,75,76,83,93]. Five studies used the culture medium supplemented with 0.64% phenol as a positive control [21,35,45,52,63]. 

It is worth noting that the assessment criteria were inconsistent. In most studies, a reduction in cell viability of more than 30% was considered a cytotoxic effect, but the threshold for cytotoxicity was at 75% in multiple studies [68,92]. In addition, many of the included studies claimed to have graded the toxicity from quantitative results in extract tests according to ISO standards [32,34,35,45,52,53,54,86,99].

#### 3.3.4. Outcome

The relationship between Zn-based BMs and cytotoxicity ranged from excellent biocompatibility to apparent cytotoxicity. Notably, the majority of the selected studies suggested that Zn-based BMs were nontoxic or produced toxic effects only under specific conditions, such as with a highly concentrated extraction solution. However, only two studies reported that pure Zn or its alloys were toxic [26,58].

## 4. Discussion

Zn-based BMs have been proposed and developed for biomedical implant materials. Alloying is a common way to improve the material properties of Zn-based BMs. This systematic review assessed the potential cytotoxic effects of Zn and its alloys. On the basis of the results, the current quality assessment of toxicity studies was assessed to be highly heterogeneous, with different study designs and non-standardized procedures making it difficult for quantitative analysis. A qualitative analysis showed that the cytotoxicity of Zn-based BMs is mainly determined by three factors: the Zn-based materials, tested cells, and test system (Figure 4).

### 4.1. Effects of the Materials on Cytotoxicity

#### 4.1.1. Material Processing

Various processing techniques were used to improve the mechanical performance of Zn-based BMs. However, these techniques also changed the material microstructure, possibly influencing their corrosion behavior and biocompatibility. The as-cast alloys suffered from significant nonuniform micro-galvanic corrosion, although the biodegradation uniformity was improved by the hot extrusion or rolling processes [20,44]. After extrusion, primary dendritic phases were broken and distributed along the extrusion direction, thereby refining the grains distinctly due to dynamical recrystallization [24]. The more uniform corrosion and reduced corrosion rate brought about by the refinement of the second phase of the bottom circulating water-cooled casting method was also demonstrated, leading to higher cell viability than conventional casting in 100% extracts [50]. Nevertheless, the opposite was true for as-extruded Zn–1.2Mg alloy, which had lower cell survival than its as-cast alloys due to a higher concentration of Zn ions resulting from a higher corrosion rate [27]. This might be related to the Mg_2_Zn_11_ phase, which is distributed relatively uniformly at grain boundaries and in the Zn matrix, involved in the formation of micro-galvanic cells [64,103]. Moreover, the corrosion resistance of the Zn matrix could be dramatically improved by appropriate heat treatment and plastic processing, resulting in a decrease in released metal ions and higher cytocompatibility [37,48]. Interestingly, one study confirmed that the crystallization process of the material affected its biocompatibility [33]. In brief, the effects of processing on the cytotoxicity of Zn-based BMs are mainly caused by changing a material’s microstructure and corrosion behavior. When designing a novel biomaterial, the relationship between the improvement of mechanical properties and the consequent change in biocompatibility has to be evaluated.

#### 4.1.2. Alloying and Its Micro-Galvanic Corrosion 

The degradation behavior of Zn-based BMs in the body is intrinsically determined by their corrosion process [104]. The corrosion behavior of metals depends on metallurgical factors such as alloy composition, phase precipitation, and segregation of alloying elements and impurities [105,106]. Therefore, the elements added into Zn-based alloys can increase their cytotoxicity by promoting micro-galvanic corrosion. 

Numerous studies demonstrated that adding Mg to pure Zn improved the biocompatibility of Zn-based alloys [23,43,82]. However, studies confirmed that the cell viability of the Zn–Mg alloys was not subject to monotonic variation with the Mg content [31,70], possibly due to the combined effect of grain refinement and passivation [82]. Copper is a component of numerous enzymes and plays a crucial role in the response to oxidative stress [107,108]. One study demonstrated that the cytocompatibility of the Zn–1Cu alloy was significantly higher than that of the pure Zn [80]. The beneficial effects of Ca, Sr, Fe, Ag, Mn, and Li as alloying elements on cytocompatibility have also been proven [55,58,69,90]. The positive effect on the biocompatibility of alloying elements was shown to be provided by Cu > Ca > Ag in decreasing order [41]. 

Alloying elements might interfere with the toxicity assessment by creating degradation reactions. As an example, the silver ions from Zn–Ag alloys could combine with chloride ions that existed in the medium to form a precipitate of silver chloride [66]. The concentration of Zn ions in the medium is decreased by alloying, but the toxic effect of the insoluble metal salts may be unknown. Excess intake of Al^3+^ is considered toxic. However, studies have reported that Zn–Al alloys have no harmful effects on HUVECs in diluted extracts, possibly because the Al^3+^ concentrations were negligible compared to the half-maximal inhibitory concentration (IC_50_) of Al^3+^ [26]. Although a high concentration of rare-earth elements (REEs) inhibited ATPase activity and caused metabolic disorders in the body [109,110], the addition of REEs to the Zn matrix also led to favorable cytocompatibility [57,68,78,94]. Undoubtedly, the release rate and content of alloying elements are essential factors influencing the cytotoxicity of Zn-based alloys.

#### 4.1.3. Surface Treatment

Pretreatment could alter the surface morphology, wettability, and roughness of the material and consequently impact its biocompatibility [111]. Among the studies included in the analysis, sample pre-cultivation was the most common pretreatment. Specifically, surface stabilization treatment resulted in a stable surface oxide film which inhibited the release of Zn^2+^, decreasing the cytotoxic effect [44,48,91]. The components of the medium used for pre-cultivation are also crucial. The presence of BSA during pre-incubation resulted in the best wettability and the lowest ion release in the initial stages of the exposure. In this case, the biocompatibility was better than that of untreated groups [72]. Li et al. investigated the impact of sterilization treatments on the cytocompatibility of Zn-based BMs. Due to the excessive release of Zn ions and a local concentration over the cellular tolerance capacity, the autoclave-treated Zn matrix exhibited apparent cytotoxic effects on fibroblasts [75]. Furthermore, cell viability in extracts of the polished-textured samples was higher than those of the fine-textured and coarse-textured samples [92]. Likewise, sandblasting treatment of the surface of Zn-based alloy specimens decreased the cell viability due to localized corrosion of the samples [93]. Therefore, as a means of changing the microstructure of tested samples, pretreatment had prominent effects on cytotoxicity.

### 4.2. Effects of Tested Cells on Cytotoxicity 

The various selected cell lines were mainly associated with the vascular and orthopedic research fields. Given that cytotoxicity is determined by the cellular tolerance of degradation products released from the Zn-based metals [108], the selection of tested cells was a vital factor in the toxicity assessment. Furthermore, metal ions usually affect cell growth in a concentration-dependent manner [60], and the tolerance of distinct cell types to metal ions is also inconsistent [76]. Hence, different cell lines could produce distinct outcomes in the toxicity assessment. For instance, Zn–Li alloy showed good biocompatibility with MC3T3-E1 cells and HUVECs [58], while its extracts showed significant cytotoxicity to L929 cells [61]. Milenin et al. determined that the viability of hDPSC was significantly higher than that of the Saos-2 under the same conditions [70]. Endothelial cells usually exhibited better cell viability compared to L929 cells, MC3T3-E1 cells, and vascular smooth muscle cells in toxicity evaluation [18,58,60], possibly because Zn, as an antioxidant and endothelial membrane stabilizer, could enhance endothelium integrity [112].

Several studies utilized stem cells with broad differentiation potential, and all these cell lines exhibited great vitality [30]. Even in the same research field, distinct cell lines have different tolerances. HOS cells showed a significant reduction in cell viability and induced cytotoxicity at a higher extract concentration, while the same conditions had a slight negative impact on the viability of MG-63 cells [27]. Also, MG-63 cells were more tolerant to Zn ions than MC3T3-E1 cells [56]. Moreover, MG-63 cells have been recommended for in vitro evaluation because they could closely simulate human cells [57]. However, this cell line may display heterogeneity among different cell populations due to donor factors. Considering the inhomogeneity of primary cells, established cell lines are more recommendable for use unless reproducibility and accuracy of the response can be demonstrated. 

### 4.3. Effects of Test System on Cytotoxicity

#### 4.3.1. Parameters of the Extract Tests

In addition to the corrosion properties of the alloy itself, the degradation rate mainly depends on the culture medium used. The ingredients of different media could be varied, and the selection of the cultivation medium is usually dictated by the cell type used. It was reported that the relatively low Zn ion release in McCoy’s 5A medium than in DMEM or DMEM/F-12 could be attributed to increased passivation film formation by a high concentration of HPO_4_^2−^ in the medium [44]. It is worth mentioning that extracts prepared by rinsing in α-MEM would help to stimulate the physiological environment [66]. Capek et al. confirmed that the ZnCl_2_ was less toxic to L929 cells in DMEM than in α-MEM, possibly since DMEM contains more glucose, amino acids, and vitamins, consequently having a strong buffering effect [76]. To provide serum proteins in simulated body fluid, adding 10% FBS to the cell culture medium is a common practice. Notably, the presence of FBS in the extraction medium could accelerate the initial corrosion process of the Zn matrix, leading to the additional potential for cytotoxicity [16]. It could be due to rapid protein adsorption on the Zn surface inhibits initial surface passivation with a protective Zn phosphate layer [113].

Among the included studies, there was significant heterogeneity in the extraction ratio of prepared extracts, partly due to different versions of the ISO standards being referred to, but more often due to the adoption of unprecedented extraction ratios. Extraction for 24 h was thought not to be sufficient to obtain an extract that represents the tested material used in practice. Hence, an immersion time of 72 h is recommended in the latest ISO standard (10993-12: 2021). Few studies adopted this latest standard. It was observed that undiluted or high-concentration extracts could exhibit toxicity effects, which could be put down to high ion concentrations and osmotic pressure inhibiting cell adhesion and growth [37]. Wang et al. suggested using a minimum of 6–10 dilutions in evaluating Mg-based BMs. This range mimics the continual clearance of absorbable ions from the circulatory system [108]. Several studies followed this recommendation and reported no toxicity effect with 10% or 12.5% extracts of Zn matrix BMs. Notably, the concentration dependence of Zn ions is more pronounced than that of Mg [114,115]. Therefore, the most appropriate dilutions for toxicity testing of Zn-based BMs to mimic the in vivo environment are yet to be demonstrated.

#### 4.3.2. Direct Contact Tests 

Most included studies have the same study endpoints in both extraction and direct contact tests. However, one study reported that tested samples had good biocompatibility in extract experiments while exhibiting cytotoxicity in the direct contact experiments [56], possibly since in vitro direct cultivation of cells on the Zn matrix was hampered by rapid degradation and partial shedding of degradation products [73]. Even though the material is nontoxic, it might interfere with the proliferation of cells to some degree [67]. It is difficult to determine the primary factors leading to decreasing cell viability and adhesion. These factors include the increase in local pH, change in surface morphology, shedding of the corrosion layer, and surface composition. In addition, cell adhesion and proliferation largely depend on extracellular matrix deposition, which is controlled by the protein adsorption capacity of a matrix surface [86].

Although direct contact tests cannot capture all complexities in vitro, they are still necessary for rapid initial screening of cytocompatibility of medical devices. 

Since tested cells can be influenced rapidly by released soluble corrosion products [110]. Since rapid protein adsorption occurs on the surface of the specimen after implantation, it is advisable to mimic this process by pre-culturing the sample in vitro when performing direct toxicity tests.

#### 4.3.3. Selected Assays 

Tetrazolium salt-based assays were used particularly frequently in toxicity assessment in the included studies. Furthermore, MTT and XTT are classical toxicity assessment methods recommended in ISO 10993 standards. However, existing studies have shown that tetrazolium-based assays can be confounded by the presence of metals, leading to false positive or negative effects [74,108]. Some authors used other parameters for toxicity assessment apart from cell viability and survival, such as the comet assay to detect the degree of DNA damage [21] and flow cytometry to evaluate the cell cycle [26]. The application of multiparametric assessment to support the observation of toxicity by a single endpoint requires more funding and research. With the development of in vitro assessments in toxicology, new paradigms of analysis, such as proteomics, genomics, and pathway analyses, contribute to the understanding of the mechanisms involved in the toxicity pathways beyond providing evidence for cell death [116]. Although these paradigms do not yet permeate the published toxicity assessments of Zn-based BMs, they might be a promising direction in the future.

#### 4.3.4. The Criteria of Cytotoxicity Evaluation

The obtained data appears to be comparable only within the results of the same study or when stringently standardized. The criteria for toxicity evaluation in quantitative and qualitative tests were different. According to the ISO-10993 standard, the toxicity results were only graded in qualitative tests. By assessing the changes in tested cells, the change from normal morphology should be graded into five levels, and the numerical grade greater than two was considered a cytotoxic effect. Only one toxicity threshold was set in the quantitative evaluation, i.e., a reduction in cell viability of more than 30% was considered a cytotoxic effect. Non-standardized evaluation criteria make the quantitative analysis of toxicity data difficult and make a direct comparison of study results impossible.

### 4.4. Strengths and Limitations of This Work

To our knowledge, this is the first systematic review of the cytotoxicity of Zn and its alloys. First, this review used the ToxRTool tool to assess the quality of toxicity evidence from the included studies, revealing the relatively lower quality of in vitro toxicity assessment. Subsequently, a detailed analysis was performed of the heterogeneity of toxicity testing in the included studies. However, this systematic review had three limitations, the first being the large volume of data from the included articles and the significant heterogeneity among the studies preventing a quantitative data analysis. Even though some trends regarding the relationship between cytotoxicity and physicochemical effects were identified, the specific influence and magnitude of each factor remain elusive. Secondly, our review was not registered in PROSPERO because RCT procedures do not apply to preclinical studies. Thirdly, this study only focused on the cytotoxicity of Zn-based BMs, while the biocompatibility is wider-ranging with many more elements than cytotoxicity. Therefore, systematic reviews regarding other aspects of biocompatibility (e.g., immunogenicity, inflammatory response, or tissue compatibility) should be performed in the future. 

## 5. Conclusions

This systematic review aimed to provide insight into the available literature exploring the cytotoxicity of biodegradable Zn and its alloys. Within the limitations of this study, the following conclusions can be drawn:High heterogeneity exists in the implementation of the included studies and the assessment results of the toxicity studies.The qualitative analysis demonstrated that biodegradable Zn and its alloys had conditionally cytotoxic effects, mainly dependent on the Zn-based materials, tested cells, and test systems.The material processing technologies and alloying elements had a potential effect on the toxicity of Zn-based BMs due to modifications in microstructure and corrosion characteristics.Endothelial cells had better tolerance to the toxic effects of Zn-based BMs than other tested cells.A standardized in vitro toxicity assessment system for biodegradable metals is still lacking, and further construction is required. In addition, researchers in this field need to comply with existing evaluation criteria and report test procedures in as much detail as possible to make the study data more informative and valuable to promote translational research and the long-term development of Zn-based BMs.

## Figures and Tables

**Figure 1 jfb-14-00206-f001:**
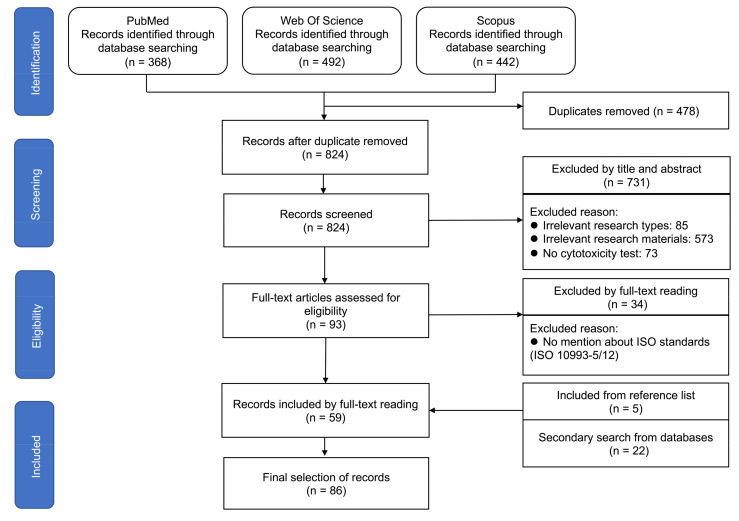
Flow diagram of the screening and selection process, according to the PRISMA statement.

**Figure 2 jfb-14-00206-f002:**
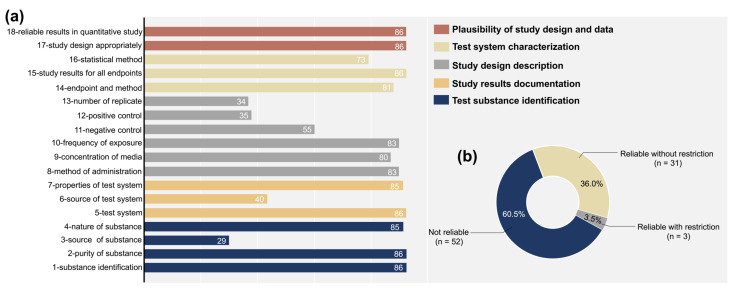
Results of quality assessment of included articles using the ToxRTool: (**a**) total score for each item of the ToxRTool in vitro criteria; (**b**) articles classified into three categories on the basis of quality evaluation results (reliable without restriction, reliable with restriction, and not reliable).

**Figure 3 jfb-14-00206-f003:**
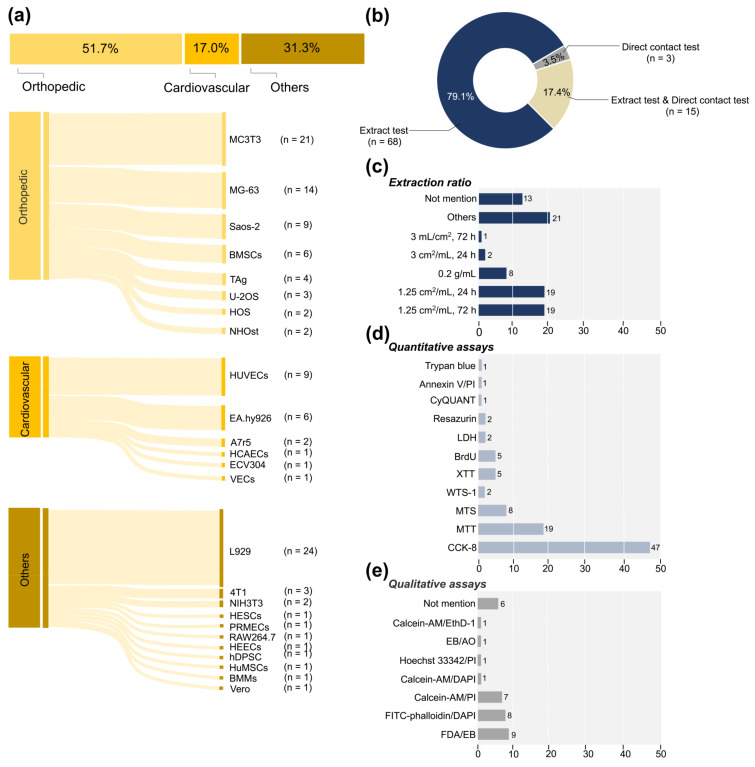
The main characteristics of the extracted experimental data: (**a**) selected cell types in studies primarily divided into three categories (cells related to cardiovascular, orthopedics, and other types); (**b**) cytotoxicity evaluation test methods; (**c**) extraction ratio of prepared extracts; (**d**,**e**) quantitative and qualitative assays used in cytotoxicity evaluation.

**Figure 4 jfb-14-00206-f004:**
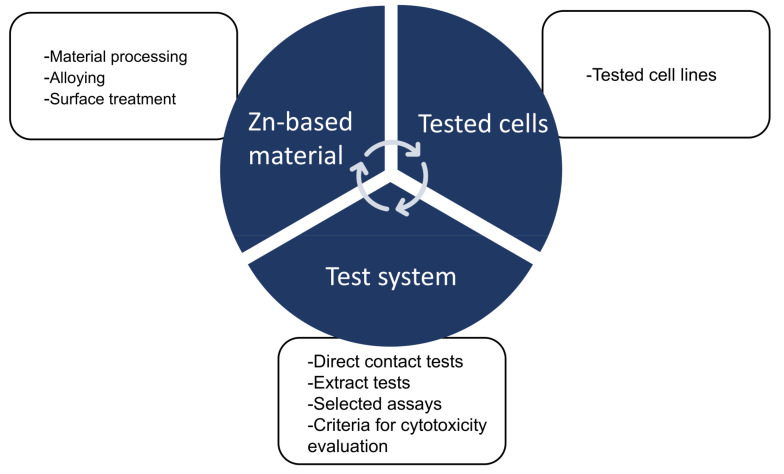
Schematic diagram of the assessment of the toxic effects of biodegradable Zn and its alloys according to three factors: Zn-based materials, tested cells, and test system.

**Table 1 jfb-14-00206-t001:** Main characteristics of the included articles.

Author /Year Ref.	Composition and Processing	Cell Line	Test Type (E, D)	Setup (SA: V, Time) and Extract Concentration (%)	Exposure Time	Negative Control	Positive Control	Assays	Others	Outcome
J. Cheng /2013 [18]	Zn (NA)	L929 ECV-304	E	1.25 cm^2^/mL, 72 h NA	E: 1, 2, 4 days	CCM	CCM with 10% DMSO	MTT	SF	Zn showed no cytotoxicity toward ECV304 cells, but could significantly reduce the cell viability of L929 cells.
M.S. Dambatta /2015 [19]	Zn Zn–3Mg (AC)	NHOst (P7)	E	0.1, 0.5, 1.0, 2.0 mg/mL, 72 h NA	E: 1, 3, 7 days	CCM	NA	MTS	Filtered	The alloy’s extract toward NHOst cells at low concentrations was cytocompatible (<0.5 mg/mL).
H. Gong /2015 [20]	Zn–1Mg (HE)	L929	E	Radio: NA, 72 h 6.25%	E: 24, 72 h	CCM	NA	MTS	SF, Filtered	Zn–1Mg alloy was biocompatible.
J. Kubásek /2015 [21]	Pure Zn Zn–0.8Mg Zn–1.6Mg (AC, HE)	U-2 OS L929 (P3)	E	NA 75%, 50%, 25%	E: 24 h	CCM with 5% FBS	CCM with 5% FBS and 0.64% phenol	WST-1		The maximum safe concentrations of Zn^2+^ for the U-2OS and L929 cells were 120 μM and 80 μM, respectively.
N.S. Murni /2015 [22]	Zn–3Mg Zn (AC)	NHOst primary cells	E	0.75 mg/mL, 72 h NA	E: 1, 3, 7 days	CCM	NA	MTS Annexin V/PI FITC–phalloidin		Zn–3Mg alloy extract exhibited adjustable cytotoxic effects on normal human osteoblast cells at the concentration of 0.75 mg/mL.
Z. Tang /2016 [23]	Zn Zn–3Cu–xMg (x = 0, 0.1, 0.5, 1.0 wt.%) (AC)	EA.hy926	E	1.25 cm^2^/mL, 72 h 10%, 50%, 100%	E: 1, 3, 5 days	NA	NA	CCK-8		Zn–3Cu–xMg alloys were biocompatible.
J. Niu /2016 [24]	Zn–4wt.%Cu (AC, HE)	EA.hy926	E	1.25 cm^2^/mL, 72 h 10%, 50%, 100%	E: 1, 3, 5 days	Ti	NA	CCK-8		Zn–4Cu presented acceptable toxicity toward human endothelial cells.
E. Jablonská /2016 [25]	Zn–1.5Mg (AC)	L929 U-2 OS	E & D	87.5 cm^2^/mL, 24 h 100%, 50%	E: 1 day D: 24 h	E: CCM D: Untreated sample	NA	WST-1 DAPI	Pre-incubation	Pre-incubation significantly increased metabolic activity of L929 in indirect test, as well as number of U-2OS cells adhered to the surface of the alloy.
C. Wang /2016 [26]	Zn ZA4-1 ZA4-3 ZA6-1 (HE)	HUVECs	E	1.25 mL/cm^2^, 24 h 100%, 50%	E: 1, 2, 4 days	CCM	NA	CCK-8	SF	Cytotoxic effect was found in 100% extracts of both pure Zn and Zn alloys, while no cytotoxicity was observed after dilution.
C. Shen /2016 [27]	Zn–1.22%Mg (AC, HE)	HOS MG-63	E	1.25 cm^2^/mL, 48 h 100%, 75%, 50%, 25%, 12.5%	E: 3 days	CCM	CCM with 5% DMSO	MTT		The as-extruded alloy had no potential cytotoxicity and tolerance in cellular applications.
G. Levy /2017 [28]	Zn–1%Mg Zn–1%Mg–0.5%Ca (AC)	Saos-2	E	1.25 cm^2^/mL, 24 h NA	E: 24, 48 h	Cells in CCM	CCM with 10% DMSO	CCK-8	Pre-incubation	The safety of all the tested zinc alloys was established in terms of their toxic effect on cells.
Z. Tang /2017 [29]	Zn Zn–xCu (x = 1, 2, 3, 4 wt.%) (AC, HE)	EA.hy926	E	1.25 cm^2^/mL, 72 h 10%, 50%, 100%	E: 1, 3, 5 days	NA	NA	CCK-8		Zn–xCu alloys were cytocompatible with human endothelial cells.
D. Zhu /2017 [30]	Pure Zn (NA)	hMSCs	E & D	1.25 mL/cm^2^, 7 days Zn ion (20−30 μM)	E: 1, 7, 14 days D: 14 days	Cells in CCM	No cells in CCM	MTT Calcein- AM		Cell motility was higher on Zn than on AZ31.
T. Ren /2018 [31]	Zn–xMg0.5Zr (x = 0.5, 1, 1.5 wt.%) Zn (AC)	L929	E	Ratio: NA, 24 h 100%, 50%, 25%	E: 1, 2, 3 days	NA	NA	MTT		The Zn–Mg–Zr alloys showed nontoxicity through in vitro cytotoxicity tests.
X. Tong /2018 [32]	Zn Zn–Ge (HE, HR)	MC3T3-E1	E	1.25 mL/cm^2^, 72 h 100%, 50%, 25%, 12.5%	E: 3 days	NA	NA	CCK-8		The <12.5% extracts of both the as-cast Zn–5Ge alloy and pure Zn showed grade 0 cytotoxicity.
N. Annonay /2018 [33]	Zn ZnZr (RF magnetron co-sputtering)	HUVECs	D	NA	E: 72 h	NA	NA	MTT Resazurin		Human endothelial cells indicated good cytocompatibility of both amorphous and crystalline films with zinc content above 80% at such thin metallic glass layers.
Y. Chen /2018 [34]	Pure Zn (AE)	PRMECs	E	1.25 cm^2^/mL, 24 h 100%, 80%, 60%, 40%, 20%	NA	Ti	NA	CCK-8 Calcein-AM/PI		100% and 80% pure zinc extracts were Grade 1, while 60%, 40%, and 20% extracts were Grade 0.
C. Xiao /2018 [35]	Zn Zn–0.05Mg (AC, AE)	L929	E	1.25 mL/cm^2^, 72 h 100%, 50%, 10%	E: 1, 3, 5 days	CCM	CCM with 0.64% phenol	MTT		Zn and Zn–0.05Mg alloy were safe for cellular applications with a cytotoxicity grade of 0–1 to L929 cells.
P. Li /2018 [36]	Zn–4.0Ag Pure Zn (AC)	L929 Saos-2	E	3 mL/cm^2^, NA 10%, 16.7%, 33.3%, 100%	E: 24, 48 h	Ti	Cu	XTT BrdU		A cytotoxic effect that decreased the viability and proliferation of L929 and Saos-2 cells was only observed in the undiluted extracts of the Zn–4Ag alloy.
X. Tong /2019 [37]	Zn–Cu foam (ED)	MC3T3-E1	E	0.2 g/mL, 72 h 100%, 50%, 25%, 12.5%	E: 1, 3, 5 days	NA	NA	CCK-8		The 100% and 50% concentrations of the extract showed clear cytotoxicity.
Y. Zhang /2019 [38]	Zn 0.5%Li (AC, HE)	BMSCs	E	Ratio: NA, 72 h 100%, 50%, 10%	NA	CCM	NA	CCK-8		The alloy was not toxic to BMSCs.
Y. Li /2019 [39]	Porous Zn (AM)	MG-63	D & E	0.2 g/mL, 72 h 10%	E: 0, 24, 48, 72 h D: 24 h	Ti	20% DMSO	MTS	Filtered	The AM porous Zn exhibited good biocompatibility in vitro.
H. Guo /2019 [40]	Pure Zn (HE, CD)	HUVECs	E	1.25 cm^2^/mL, 24 h 100%, 50%, 10%	E: 1, 3, 5 days	CCM	NA	CCK-8	SF	The ф 0.3 mm pure Zn wire presented benign cytocompatibility in 100% concentration extract, whereas the ф 3.0 mm pure Zn wire exhibited higher cytotoxicity in 100% concentration extract.
Z. Shi /2019 [41]	Zn–0.8Mn Zn–0.8Mn–0.4X (X = Ag, Cu, Ca) (AC)	L929	E	0.2 g/mL 100%, 80%, 60%, 40% 20%	E: 48 h	100% HDPE extract	The medium with 10% FBS and 10% DMSO	MTT		The addition of Cu or Ca obviously alleviated the cytotoxic potential of Zn-0.8Mn alloy.
P. Li /2019 [42]	Zn–4Ag Zn (AC)	TAg	E	1.25 cm^2^/mL, NA 100%, 50%, 25%, 10%, 5%, 2%	E: 2, 6, 12 days	CCM and in osteogenic media	NA	CCK-8		Compared with pure Zn, the Zn–4Ag alloy seemed to exhibit no adverse cytotoxic effects on TAg cells.
P. Li /2019 [16]	Zn Zn–4Ag Zn–2Ag–1.8Au–0.2V (NA)	L929 Saos-2	E	3 cm^2^/mL, NA NA	E: 24 h	Ti	Cu	FDA/EB CCK-8		Decreased cytotoxicity was observed in the extract media without FBS.
S. Lin /2019 [43]	Pure Zn Zn–0.02Mg (AC)	HUVECs	E	1.25 cm^2^/mL, 72 h NA	E: 1, 3 days	NA	NA	MTS		Zn–0.02Mg alloy extracts promoted HUVEC activity after 1 and 3 days of incubation.
P. Li /2019 [44]	Zn Zn–xCu (x = 1, 2, 4 wt.%) (AC)	L929 Tag Saos-2	E	1.25 cm^2^/mL, 24 h NA	E: 24 h	Ti	Cu	FDA/EB CCK-8 BrdU		As-rolled Zn–4Cu alloy exhibited no apparent cytotoxic effect toward L929, TAg, or Saos-2 cells.
Y. Zhang /2019 [45]	Zn–0.8%Li Zn–0.8%Li–0.2%X (X = Li, Ag) (AC)	L929 BMSCs	E	Ratio: NA, 72 h 100%, 50%, 10%	E: 1, 3, 5 days	CCM	CCM containing 0.64% phenol	CCK-8		The cytotoxicity of these extracts of Zn–Li–Ag alloy was of Grade 0–1.
D. Zhu /2019 [46]	Pure Zn Zn−1.5%Sr Zn−1.5%Mg (AC, HR, AE)	HCAECs (P4-6 HOBs hMSCs	D & E	NA 10%, 25%, 50%	E: 5 days D: 5 days	CCM	NA	MTT CyQUANT		The measured cell viability and proliferation of three different human primary cells fared better for Zn biomaterials than AZ31.
C. Shuai /2020 [47]	Zn–Al Zn–Al–2Sn (SLM)	MG-63	D & E	1.25 cm^2^/mL, 72 h NA	E: 1, 3, 5 days D: 24 h	NA	NA	CCK-8		Zn–Al–2Sn alloy had acceptable cytocompatibility.
C. Chen /2020 [48]	Zn–1.5Cu–1.5Ag Zn (AC, AE)	EA.hy926	E	1.25 cm^2^/mL, 72 h 20%, 50%	E: 1, 2, 3 days	NA	NA	CCK-8		The as-extruded alloy exhibited good biocompatibility at cellular level.
O. Avior /2020 [49]	Zn Zn–2%Fe Zn–2%Fe–xCa (x = 0.3, 0.6, 1, 1.6 wt.%) (AC)	4T1	E	1.25 cm^2^/mL, 24 h NA	E: 24, 48 h	CCM	CCM with 90% DMEM and 10% DMSO	XTT	Filtered	All the tested alloys can be noncytotoxic substances regarding 4T1 cells.
Z. Zhang /2020 [50]	Zn–0.3Fe (AC, BCWC)	HUVECs	E	1/3 mL/cm^2^, 24 h 25%–100%	E: 24 h	CCM	NA	CCK-8		Both the alloys exhibited no cytotoxicity.
B. Jia /2020 [51]	Pure Zn Zn–xMn (x = 0.1, 0.4, 0.8 wt.%) (AE)	MC3T3-E1	E	1.25 mL/cm^2^, 24 h 25%, 50%	CCK-8: 1, 3, 5, 7 days Live/dead: 3 days	NA	NA	CCK-8 DAPI/FITC–phalloidin Live/dead	Filtered	The addition of Mn significantly improved the cytocompatibility properties of pure Zn.
X. Xu /2020 [52]	Zn–0.8Li–0.2Ag (HR)	BMSCs	E	20 mL/cm^2^ 100%, 50%,10%	E: 1, 3, 5 days	CCM	CCM with 0.64% phenol	CCK-8		Zn–0.8Li–0.2Ag alloy showed no toxicity toward BMSCs in cytotoxicity test.
Y. Li /2020 [53]	Porous Zn (AM)	MG-63	D & E	0.2 g/mL, 72 h 10%	E: 0, 24, 48, 72 h D: 24 h	Ti	20% DMSO	MTS Live/dead	Filtered	The AM porous Zn exhibited good biocompatibility in vitro.
K. Wang /2020 [54]	Zn–xTi (x = 0.05, 0.1, 0.2, 0.3 wt.%) (AC, HR)	MG-63	E	1.25 cm^2^/mL, 3 days 100%, 25%, 12.5%	E: 1 day	NA	NA	CCK-8		The extracts of both AC and HR Zn–xTi alloys at concentrations of ≤25% showed no cytotoxicity toward MG-63 cells.
J. Lin /2020 [55]	Zn–3Ge Zn–3Ge–0.5X (X = Cu, Mg, Fe) (AC, HR)	MG-63	E	1.25 cm^2^/mL, 3 days 100%, 25%, 12.5%	E: 5 days	NA	NA	CCK-8		The cell viability of MG-63 cells in the extracts of all the Zn alloys at a concentration of 12.5% exceeded 90%.
J. Lin /2020 [56]	Zn–1Cu–0.1Ti Pure Zn (AC)	MC3T3-E1 MG-63	D & E	1.25 cm^2^/mL, 72 h 100%, 25%, 12.5%	D: 24, 48 h E: 1, 3, 5 days	NA	NA	CCK-8		The extract of AC Zn–1Cu–0.1Ti alloy at a concentration ≤25% showed no significant cytotoxicity toward MC3T3-E1 and MG-63 cells.
X. Tong /2020 [57]	Zn–1Mg Zn–1Mg–0.1RE (RE = Er, Dy, Ho) (AC, HR)	MC3T3-E1 MG-63	D & E	1.25 cm^2^/mL, 72 h 100%, 25%, 12.5%	E: 1, 3, 5 days D: 24, 48 h	CCM	NA	CCK-8		The 12.5% concentration extracts of the HR Zn–1Mg and Zn–1Mg–0.1RE alloys showed good cell proliferation and growth of MG-63 without cytotoxicity.
H. Yang /2020 [58]	Zn–xMg Zn–xCa Zn–xSr Zn–xLi Zn–xMn Zn–xFe Zn–xCu Zn–xAg Zn (HE)	MC3T3-E1 HUVEC	D & E	1.25 mL/cm^2^, 24 h 100%, 50%	E: 1, 2, 4 days D: 12 h	CCM	CCM with 10% DMSO	CCK-8 DAPI/FITC–phalloidin	SF	E: Pure Zn and other binary Zn alloys exhibited severe cytotoxicity except for Zn–0.8Ca and Zn–0.1Sr. D: MC3T3-E1 cell displayed a round and unhealthy shape on materials with good cytocompatibility.
P. Li /2020 [59]	Zn–2Ag–1.8Au–0.2V (AC)	L929 Saos-2	E	3 cm^2^/mL, 24 h 33.3%, 16.7%, 10%	E: 24 h	Ti	Cu	XTT FDA/EB BrdU		It showed acceptable toxicity in the results obtained with cells exposed to 10% and 16.7% extracts and notable toxic effects in undiluted extracts.
R. Yue /2020 [60]	Zn Zn–3Cu Zn–3Cu–0.2Fe Zn–3Cu–0.5Fe (AE)	EA.hy926 A7r5	D & E	1.25 cm^2^/mL, 3 days 10%, 50%, 75%, 100%	E: 3 days D: 12 h	No cells in CCM	Cells in CCM	CCK-8 LDH Live/Dead	Filtered	EA.hy926 cells were more tolerant than A7r5 cells to the extracts of Zn–3Cu–xFe alloys.
Z. Li /2020 [61]	Zn Zn–xLi (x = 0.2–1.4 wt.%) (AC)	L929	E	0.2 g/mL, 24 h 10%, 20%, 40%, 60%, 80%, 100%	E: 1 day	CCM	DMEM with 15% DMSO	MTT		The 10% extracts of Zn–Li alloys exhibited no cytotoxicity.
H. Guo /2020 [62]	Pure Zn (LC)	MC3T3-E1	E	1.25 cm^2^/mL, 24 h 10%, 50%, 100%	E: 1, 3, 5 days	CCM	CCM with 10% DMSO	Calcein-AM/PI CCK-8	SF	Pure zinc membrane with 300 μm pores displayed acceptable MC3T3-E1 cytocompatibility in vitro.
C. Xiao /2020 [63]	Zn–0.05Mg–xAg (x = 0.5, 1.0 wt.%) (AC)	L929	E	3 mL/cm^2^, 72 h 100%, 50%, 10%	E: 1, 3, 5 days	CCM	CCM with 0.64% phenol	MTT		L929 cells grew normally after culturing for 1, 3, and 5 days in the extracts of the alloys.
L. Deng /2021 [64]	Zn–0.45Li Zn–2Li (AC, AE, AD)	L929	E	NA 25%, 100%	E: 24, 48, 72 h	NA	NA	MTT DAPI/FITC–phalloidin		The MTT cytotoxicity assay suggested a low corrosion rate and good cytocompatibility of the Zn–0.45Li alloys.
B. Jia /2021 [65]	Pure Zn Zn–xSr (x = 0, 0.1, 0.4, 0.8 wt.%) (HE)	MC3T3-E1	E	1.25 mL/cm^2^, 24 h 50%, 25%	E: 1, 3, 5, 7 days	CCM	NA	CCK-8 Live/dead DAPI/FITC–phalloidin	Filtered	Pure Zn was mildly cytotoxic to MC3T3-E1 cells but Zn–Sr alloys could significantly improve cytocompatibility.
H. Wu /2021 [66]	Pure Zn Zn–Ag Zn–Mg–Ag (AC)	MC3T3	E	Ratio: NA 12.5%	E: 24, 48, 72, 96 h	NA	NA	CCK-8 DAPI/FITC–phalloidin		The Zn–0.04Mg–2Ag porous scaffold had excellent mechanical properties and biocompatibility.
E. Farabi /2021 [67]	Zn–Al–Li (AC, AE)	HuMSCs L929	E	2 mL, 21 days 50%, 100%	E: 3 h	NA	NA	MTS		The developed Zn–4Al–0.6Li and Zn–6Al–0.4Li alloys appeared to be cytocompatible with HuMSCs and L929 cells.
Y. Yang /2021 [68]	Zn Zn–xCe (x = 1, 2, 3 wt.%) (LPBF)	MG-63	E	1.25 cm^2^/mL, 72 h NA	E: 1, 3, 7 days	CCM	NA	Calcein-AM CCK-8		Zn–Ce exhibited no obvious cell cytotoxicity.
X. Qu /2021 [69]	Pure Zn Zn–xAg (x = 0.5, 1, 2 wt.%) (HE)	MC3T3-K BMMs	E	1.25 cm^2^/mL, 24 h 50%, 33.3%, 25%, 20%	E: 24, 72 h	NA	NA	CCK-8	Filtered	Zn–2Ag alloy significantly inhibited osteoclastic differentiation of BMMs cells in vitro.
A. Milenin /2021 [70]	Zn Zn–Mg (Properzi method)	hDPSC Saos-2	E	0.2 g/mL; 0.04 g/mL, NA NA	E: 24 h	TCP	NA	MTS		Mg content of 0.0026 wt.% in the Zn-based wire provided extracts that are toxic to cancer cells and nontoxic to healthy cells.
J. Lin /2021 [71]	Zn–3Cu Zn–3Cu–0.2Ti (AC, HR, CR)	MG-63	E	1.25 cm^2^/mL, 3 days 100%, 25%, 12.5%	E: 1 day	NA	NA	CCK-8		The extracts of both HR + CR Zn–3Cu and Zn–3Cu–0.2Ti alloys at a concentration of ≤25% showed no cytotoxicity toward MG-63 cells, and the Zn–3Cu–0.2Ti alloy exhibited higher cytocompatibility than Zn–3Cu.
J. Pinc /2021 [72]	Zn–0.8Mg–0.2Sr (HE)	NIH 3T3	E	1 mL/cm^2^, 24 h 33.3%, 6.67%	E: 24, 48 h	Cells in CCM	NA	MTT Trypan blue	Pre-incubation	Poor cell viability in sample eluates was caused by the high Zn^2+^ ion release.
E. Jablonská /2021 [15]	Zn–0.8Mg (SPS)	U-2 OS L929 (P3-P20)	E	87.5 mm^2^/mL, 24 h NA	E: 24 h	NA	NA	Resazurin	5%, 10%, or without FBS	The type of medium, the concentration of FBS, mode of exposition, and cell type all influenced the cytotoxicity of the extracts.
W. Zhang /2021 [73]	Zn Zn–0.5%Cu–xFe (x = 0, 0.1, 0.2, 0.4 wt.%) (AC)	L929 Saos-2 TAg	E & D	1.25 cm^2^/mL, 24 h 100%	E: 24 h D: NA	Ti/CCM	Cu	CCK-8 FDA/EB		The extracts of Zn–0.5Cu–Fe (0.2 wt.%) alloys showed no cytotoxic effects toward tested cells.
P. Zhu /2021 [74]	Pure Zn (NA)	L929	D	-	D: 24 h	Ti	Cu	FDA/EB XTT BrdU	Pre-incubation	The direct cells cultured on Zn-based surfaces led to apparent misleading cytotoxicity with the CCK-8 assay.
P. Li /2021 [75]	Pure Zn Zn–3Cu (AC)	L929	E	1.25 cm^2^/mL, 24 h NA	E: 24 h	Ti	Cu	FDA/EB CCK-8		The extract test indicated that gamma irradiation or H_2_O_2_ gas plasma sterilization did not induce cytotoxic effects toward L929 fibroblasts on Zn and Zn–Cu alloy.
J. Capek /2021 [76]	Zn–0.8Mg–0.2Sr Zn (AC, HE)	L929 Saos-2 Tag	E	1.25 cm^2^/mL, 24 h 100%, 50%, 25%	E: 24 h	Ti	Cu	FDA/EB CCK-8 BrdU		The 25% extracts of the Zn–0.8Mg–0.2Sr alloys had no apparent adverse effects on the cell viability and proliferation of L929, Tag, and Saos-2 cells.
O. Avior /2022 [77]	Zn–2%Fe–0.6%Ca (AC)	4T1	D	1.25 mL/cm^2^, 24 h NA	D: 24 h, 48 h	Ti	NA	Live/dead	Pre-incubation	The tested alloy was suitable for cell growth under in vitro conditions, as seeded cells were adherent and viable on the alloy surface.
X. Tong /2022 [78]	Zn–1Mg–xGd (x = 0.1, 0.2, 0.3 wt.%) (AC, HR)	MG-63	E	1.25 cm^2^/mL, 3 days 12.5%, 25%, 50%, 100%	E: 24 h	NA	NA	CCK-8		High-concentration (≥50%) extracts of Zn–1Mg–0.3Gd had clear inhibitory effects on MG-63 cells.
J. Jiang /2022 [79]	Pure Zn Zn–2.2wt.% Cu–xMn (x = 0, 0.4, 0.7, 1 wt.%) (AC)	EA.hy926 A7r5	E	1.25 cm^2^/mL, 3 days 100%, 50%, 10%	E: 1, 3 days	No cells in CCM	Cells in CCM	CCK-8		Zn–2.2Cu–0.4Mn alloy exhibited acceptable in vitro cytocompatibility, comparable with pure Zn.
G. Bao /2022 [80]	Zn Zn–0.5Cu Zn–1Cu (NA)	HEECs HESCs	E	1.25 cm^2^/mL, 24 h 100%, 50%, 10%	E: 1, 3, 5 days	CCM	CCM with 10% DMSO	CCK-8	SF	The Zn–0.5Cu exhibited slightly higher-level cell viability than Cu, however, it was much lower than pure Zn and Zu–1Cu.
H. Ren /2022 [81]	Zn Porous Zn–xCu (x = 0, 1, 2, 3) (APIM)	MC3T3-E1 L929	E	Ratio: NA, 24 h NA	E: 1, 2 days	NA	NA	MTT		The alloy exhibited good cytocompatibility at a low extract concentration.
Y. Qin /2022 [82]	Zn–xMg (x = 1, 2, 5 wt.%) (AC)	MC3T3-E1	E	1.25 cm^2^/mL, 24 h 100%, 50%, 10%	E: 1, 3, 5 days	CCM	CCM with 10% DMSO	CCK-8 Calcein AM/PI		The cell viability increased with increasing Mg content.
Y. Xu /2022 [83]	Zn–0.5Cu–0.2Fe Zn (AC)	HUVEC RAW264.7 MC3T3-E1	E	1.25 cm^2^/mL, 72 h 50, 25, 12.5%	E: 24 h	Ti	Cu	LDH FDA/EB		The hot extruded Zn–Cu–Fe alloy exhibited good performance in terms of cytocompatibility.
Y. Zeng /2022 [84]	Zn–Fe–Si (AC)	HUVEC	E	1.25 cm^2^/mL, 72 h 6.25%	E: 24, 72 h	CCM	NA	MTT		The biocompatibility of the test alloy was acceptable.
Y. Liu /2022 [85]	Zn–0.5Fe (AS)	MC3T3-E1	E	6 cm^2^/mL, 24 h 12.5%, 25%, 50%	E: 1, 3, 5 days	CCM	NA	CCK-8		The Zn–0.5Fe alloy membrane had adequate biocompatibility.
D. Palai /2022 [86]	Zn Zn–xCu (x = 1, 2, 3 wt.%) (AC)	3T3 fibroblasts	E	Ratio: NA, 72 h 50%	E: 1,3, 5 days	NA	NA	MTT		The Zn–2Cu and Zn–3Cu alloys exhibited better cytocompatibility compared to pure Zn.
N.A. Gopal /2022 [87]	Zn–Ti–Cu–Ca–P (AS)	Vero cell	E	NA	E: 24, 48, 72 h	NA	NA	MTT EB/AO		The presented material can be used as a bio-implant.
N. Yang /2022 [88]	Zn–Cu–Ca Zn (AC, HR)	HUVEC L929	E	1.25 cm^2^/mL, 24 h 100%, 50%, 25%, 12.5%	E: 1, 3 days	Cell in CCM	NA	CCK-8		The alloys had good cytocompatibility for the tested cell lines.
J. Duan /2022 [89]	Zn–2Cu–0.2Mn–xLi (x = 0, 0.1, 0.38 wt.%) (AC, HE)	MC3T3-E1		1.25 mL/cm^2^, 24 h 100%, 50%, 25%	E: 1, 2, 3 days	CCM	NA	CCK-8 Calcein-AM/EthD-1 DAPI/FITC–phalloidin		MC3T3-E1 cells exhibited over 95% viability in the 25% extracts of all as-extruded alloys.
X. Zhu /2022 [90]	Zn–Mn Pure Zn (AC, HE)	L929	E	1.5 cm^2^/mL, 24 h NA	E: 24, 48, 72 h	NA	NA	MTT		The concentration of Zn^2+^ in the 100% concentration extract exceeded the safety threshold, causing the relative growth rate of cells to be lower than 100%.
G.K. Levy /2019 [91]	Zn–1Mg Zn–1Mg–0.5Ca (DC)	MSCs	D&E	Ratio: NA, 24 h NA	E: 24, 48, 72 h D: 24 h	CCM	10% DMSO	CCK-8 Live/Dead		A short and simple 1 day surface stabilization treatment in cell growth medium significantly improved cell adhesion and viability.
I. Cockerill /2019 [92]	Zn (AC)	MC3T3-E1	D&E	1.25 cm^2^/mL, 72 h 10%	D: 24 h E: 1, 3, 5 days	CCM	NA	D: SEM E: MTT		The textured Zn samples supported the adhesion of pre-osteoblasts that exhibited flat morphologies with numerous cytoplasmic extensions, and cytocompatibility tests showed >75% cell viability in 10% extracts.
P. Li/2020 [93]	Pure Zn Zn–4Ag Zn–2Ag–1.8Au–0.2V (AC)	Saos-2	E	1.25 cm^2^/mL, 24 h NA	E: 24 h	Ti	Cu	FDA/EB CCK-8		Samples treated with 250 µm sandblasting particles caused a mean decrease in viability below 70% of the control, i.e., classified as an apparent cytotoxic effect.
X. Tong /2022 [94]	Zn–xDy (x = 1, 3, 5 wt.%) Zn HR	MC3T3-E1	E	1.25 cm^2^/mL, 48 h 100%, 25%, 12.5%	E: 3 days	CCM	NA	CCK-8 Calcein-AM/PI		The HR Zn–3Dy extract with 12.5% concentration showed the highest cell viability of ∼102.1% toward MC3T3-E1 cells among all samples tested.
M. Wątroba /2022 [95]	Zn Zn–3Ag Zn–3Ag–0.5Mg (AC)	MG-63	E&D	1.25 cm^2^/mL, 24 h 100%, 50%, 25%, 12.5%, 5%	E: 24 h D: 24 h	NA	CCM	WST-8 LDH Calcein-AM/DAPI		Cytotoxicity tests showed almost no significant differences between pure Zn and Zn alloys.
T. Di/2022 [96]	Zn–1Cu–xAg (x = 0.5, 1 wt.%) (HE)	MC3T3-E1	E&D	Ratio: NA, 24 h 100%, 50%, 25%, 12.5%, 6.25%	E: 1, 2, 3 days	CCM	NA	Hoechst 33342/PI MTT		The cytotoxicity grade of the twofold diluted extracts of Zn–1Cu–xAg alloy was 0–1, and the cytocompatibility met the requirements for orthopedic application.
Z. Wang /2022 [97]	Pure Zn Zn–Mg (HE)	MC3T3-E1 VEC	E	1.25 cm^2^/mL, 24 h NA	E: 1, 3, 5 days	NA	NA	CCK-8		Zn–Mg alloys examined in this study exhibited good cytocompatibility in vitro with osteoblasts and endothelial cells.
L.B. Tzion-Mottye /2022 [98]	Zn–2%Fe (AC)	Mus musculus 4T1	E	1.25 cm^2^/mL, 24 h 10%	E: 24 h, 48 h	Ti	NA	XTT		Indirect cell viability assessment showed that the addition of Mn tended to increase cell viability in vitro.
Z. Zhang /2022 [99]	Zn–0.60Mn–0.064Mg Zn–0.81Mn–0.049Mg (HE)	MC3T3-E1	E	1.25 cm^2^/mL, NA 100%, 25%–75%	E: 1, 3 days	CCM	Cells in CCM	CCK-8 Live/dead		Both alloys had biocompatibility.
L. Sheng /2022 [100]	Zn–1.5Fe (SPS)	MG-63	E	1.25 cm^2^/mL, 24 h NA	E: 3, 5, 7 days	NA	NA	CCK-8		The viability of MG-63 on Zn–1.5Fe alloys was over 85%.
L. Jin /2022 [101]	Zn Zn–0.5Li (HR)	MC3T3-E1	E	1.25 cm^2^/mL, 24 h 100%, 50%, 25%	E: 1, 3, 5 days	CCM	NA	CCK-8 Calcein-AM/PI	SF	The biocompatibility of Zn–0.5Li was higher than that of pure Zn.

Abbreviations: not available (NA); reference (Ref.); extract test (E); direct contact test (D); as-casting (AC); as-extruded (AE); hot-extruded (HE); hot-rolling (HR); bottom circulating water-cooled casting (BCWC); high-pressure solidification (HPS); laser cutting technology (LC); die-casting (DC); air pressure infiltration method (APIM); cold-drawing (CD); electro-deposition (ED); additively manufactured (AM); selective laser melting (SLM); hot-treatment (HT); laser powder bed fusion (LPBF); cells cultured in tissue culture plates (TCP); high-pressure solidification (HPS); spark plasma sintering (SPS); as-sintered (AS); cell culture medium (CCM); withdrawn supernatant fluid (SF); Ti–6Al–4V (Ti).

## Data Availability

Experimental data from this study are available from the corresponding author upon reasonable request.

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
