# Peer review of "Cytotoxicity of Biodegradable Zinc and Its Alloys: A Systematic Review"

_jfb, 2023, doi:10.3390/jfb14040206_

Round 1

Reviewer 1 Report

In this systematic review, the authors aim to investigate the cytotoxicity of biodegradable zinc and its alloys. The article provides a comprehensive overview of the current state of knowledge on this topic, synthesizing data from a variety of studies and presenting the findings in a clear and organized manner. Overall, I believe this article is a valuable contribution to the field of biomaterials and biocompatibility.

One of the strengths of this article is its thoroughness. The authors conducted a systematic search of the literature, identified relevant studies, and critically appraised each study to determine its relevance and quality. The inclusion criteria were well-defined, and the authors provided detailed descriptions of each study's methods, results, and limitations. This level of rigor adds credibility to the article's findings and makes it a useful resource for other researchers in the field.

Another strength of this article is its clarity. The authors organized the information in a logical and easy-to-follow manner, with clear headings and subheadings. The language used was concise and technical terms were defined or explained, making the article accessible to a broad audience. The inclusion of tables and figures also helped to summarize the findings and enhance their visual impact.

However, there are a few areas where the article could be improved. The authors focused solely on cytotoxicity and did not consider other aspects of biocompatibility, such as immunogenicity, inflammatory response, or tissue compatibility. While cytotoxicity is an important factor to consider, it is not the only determinant of a material's biocompatibility. Therefore, a more comprehensive assessment of biocompatibility would have added value to the article. This should be a topic of future studies.

In conclusion, the article "Cytotoxicity of biodegradable zinc and its alloys: A systematic review" provides a comprehensive overview of the current state of knowledge on the cytotoxicity of biodegradable zinc and its alloys. The article's thoroughness and clarity make it a valuable resource for researchers in the field of biomaterials and biocompatibility. This article is therefore recommend for publication.

Author Response

Comments

Point 1: In this systematic review, the authors aim to investigate the cytotoxicity of biodegradable zinc and its alloys. The article provides a comprehensive overview of the current state of knowledge on this topic, synthesizing data from a variety of studies and presenting the findings in a clear and organized manner. Overall, I believe this article is a valuable contribution to the field of biomaterials and biocompatibility.

One of the strengths of this article is its thoroughness. The authors conducted a systematic search of the literature, identified relevant studies, and critically appraised each study to determine its relevance and quality. The inclusion criteria were well-defined, and the authors provided detailed descriptions of each study's methods, results, and limitations. This level of rigor adds credibility to the article's findings and makes it a useful resource for other researchers in the field.

Another strength of this article is its clarity. The authors organized the information in a logical and easy-to-follow manner, with clear headings and subheadings. The language used was concise and technical terms were defined or explained, making the article accessible to a broad audience. The inclusion of tables and figures also helped to summarize the findings and enhance their visual impact.

However, there are a few areas where the article could be improved. The authors focused solely on cytotoxicity and did not consider other aspects of biocompatibility, such as immunogenicity, inflammatory response, or tissue compatibility. While cytotoxicity is an important factor to consider, it is not the only determinant of a material's biocompatibility. Therefore, a more comprehensive assessment of biocompatibility would have added value to the article. This should be a topic of future studies.

In conclusion, the article "Cytotoxicity of biodegradable zinc and its alloys: A systematic review" provides a comprehensive overview of the current state of knowledge on the cytotoxicity of biodegradable zinc and its alloys. The article's thoroughness and clarity make it a valuable resource for researchers in the field of biomaterials and biocompatibility. This article is therefore recommend for publication.

Response 1: We are highly grateful for the reviewer's positive comments. As mentioned by the reviewer, the concept of biocompatibility is wide-ranging and requires consideration of not only cytotoxicity but also immunogenicity, inflammatory response, and tissue compatibility. However, we did focus solely on cytotoxicity. There are two main reasons. Firstly, we considered the cytotoxicity test is the first step in evaluating the biocompatibility of biodegradable Zn-based metals. Exploring other biocompatibility aspects is generally further developed and studied based on demonstrating the absence of cytotoxic effects of the zinc-based metals. Secondly, biodegradable zinc-based material is still in its basic research stage, and the majority of biocompatibility studies were focused on cytotoxicity. Thus, there are not yet enough studies for the systematic evaluation of other aspects of biocompatibility. With the development of the biodegradable zinc-based material field in the future, we believe a comprehensive evaluation of biocompatibility for zinc-based metals could be done, which is what we want to do in the following work. 

Based on this, we added the discussion of the limitations of our work "Systematic review regarding other biocompatibility (e.g., immunogenicity, inflammatory response, or tissue compatibility) should be done in the future" on page 26, lines 437-440 in the revised manuscript. We thank you again for your precious suggestions! 

Reviewer 2 Report

Review Notes for “Cytotoxicity of biodegradable zinc and its alloys: A systematic review”

This review is well written and will be greatly appreciated by researchers working in biodegradable metals. The format is very strong and elegant for selecting the articles to discuss. That is a valuable resource alone, identifying the articles that meet strong criteria of performance and quality. The conclusions, unfortunately, point to what was becoming evident in the field, that there are so many variables that it is not possible yet to settle on one assay system, cell type or culture medium. Or compare across cell types even. It may never be possible. But this is an excellent beginning to understand the whole picture and the many variables.

Minor concerns:

Page 21, line 206: Suggest using “bovine serum albumin” instead of “bull serum albumin”, as there is no way to tell if the albumin is from a male bull or a female cow.

Page 21, line 225 and in Table 3d: The authors state: “In five studies, FITC/DAPI dye was used for cytoskeletal staining to observe cell morphology” (refs). However, I believe that they are referring to FITC-phalloidin/DAPI dyes instead. FITC is just a fluorescent molecule and cannot label anything. If phalloidin is labeled covalently with FITC (or any other fluorophore), then the phalloidin binds strongly to actin microtubules and the cellular cytoskeleton can be seen. DAPI is a nuclear stain, labeling nuclei. To say that FITC/DAPI was used for cytoskeletal staining is incorrect. FITC-phalloidin is used for cytoskeletal staining, not DAPI. They are used together to identify the cell by the nucleus and then examine the cytoskeleton with the labeled phalloidin.

Page 21, lines 230-232. The authors state: “The unexposed group and the cell culture medium supplemented with dimethyl sulfoxide (DMSO) were usually chosen as negative and positive control groups, respectively.” This is all relative to how much DMSO is added. I am assuming that negative and positive refer to no cytotoxicity versus extensive cytotoxicity (all alive vs all dead)? That should be clarified. Then, for many of the assay kits, the dye is diluted in a very small amount of DMSO, so the negative control may be cell culture medium supplemented with the same small amount of DMSO (no cell death). Then, the medium supplemented with a large amount of DMSO, which leads to complete cell death, is the “positive” control. So a suggested different wording might be: The unexposed group with cell medium alone or with a small amount of DMSO (to control for additive diluents) is the viable (negative cytotoxicity) control group, while unexposed cells with a large amount of DMSO is the nonviable (positive) control group.

Author Response

Comments

This review is well written and will be greatly appreciated by researchers working in biodegradable metals. The format is very strong and elegant for selecting the articles to discuss. That is a valuable resource alone, identifying the articles that meet strong criteria of performance and quality. The conclusions, unfortunately, point to what was becoming evident in the field, that there are so many variables that it is not possible yet to settle on one assay system, cell type or culture medium. Or compare across cell types even. It may never be possible. But this is an excellent beginning to understand the whole picture and the many variables.

Minor concerns:

Point 1: Page 21, line 206: Suggest using “bovine serum albumin” instead of “bull serum albumin”, as there is no way to tell if the albumin is from a male bull or a female cow.

Response 1: We sincerely thank the reviewer for careful reading and pointing out drawbacks in our manuscript. We have revised the “bull serum albumin” into “bovine serum albumin” in the revised manuscript (Page 21 line 204).

Point 2: Page 21, line 225 and in Table 3d: The authors state: “In five studies, FITC/DAPI dye was used for cytoskeletal staining to observe cell morphology” (refs). However, I believe that they are referring to FITC-phalloidin/DAPI dyes instead. FITC is just a fluorescent molecule and cannot label anything. If phalloidin is labeled covalently with FITC (or any other fluorophore), then the phalloidin binds strongly to actin microtubules and the cellular cytoskeleton can be seen. DAPI is a nuclear stain, labeling nuclei. To say that FITC/DAPI was used for cytoskeletal staining is incorrect. FITC-phalloidin is used for cytoskeletal staining, not DAPI. They are used together to identify the cell by the nucleus and then examine the cytoskeleton with the labeled phalloidin.

Response 2: Thanks for pinpointing the mistake. In the resubmitted manuscript, we revised FITC/DAPI to FITC-phalloidin/DAPI in Page 21 line 223, Table 1, and Figure 3e.

Point 3: Page 21, lines 230-232. The authors state: “The unexposed group and the cell culture medium supplemented with dimethyl sulfoxide (DMSO) were usually chosen as negative and positive control groups, respectively.” This is all relative to how much DMSO is added. I am assuming that negative and positive refer to no cytotoxicity versus extensive cytotoxicity (all alive vs all dead)? That should be clarified. Then, for many of the assay kits, the dye is diluted in a very small amount of DMSO, so the negative control may be cell culture medium supplemented with the same small amount of DMSO (no cell death). Then, the medium supplemented with a large amount of DMSO, which leads to complete cell death, is the “positive” control. So a suggested different wording might be: The unexposed group with cell medium alone or with a small amount of DMSO (to control for additive diluents) is the viable (negative cytotoxicity) control group, while unexposed cells with a large amount of DMSO is the nonviable (positive) control group.

Response 3: We so much appreciated the reviewer's suggestion. Among the included studies, the test cell in the cell culture medium was usually set as the negative control, compared to experimental groups added with different concentrations of biodegradable zinc-based metal extracts. Added 5-20% DMSO to the cell culture medium were set as positive control groups in some studies. As the reviewer's suggestion, the amount of DMSO added should be clearly stated in the first manuscript. In the revised manuscript, we have indicated the DMSO content in the positive control groups.

Regarding your comment that many staining kits also add very small amounts of DMSO, and suggested using "The unexposed group with cell medium alone or with a small amount of DMSO." In the post-treatment incubation phase, it examines the effect of the treatment factor (different concentrations of zinc-based BMs extracts) on the cytotoxicity, where the negative and positive control were set up to exclude false positive and false negative, respectively. In the test phase, since the staining reagent is added to each group (including experimental groups and control groups), even though very small amounts of DMSO are present in the staining reagent. We thought that would not be a specific treatment for the negative control group. Besides, the presence of small DMSO content could be ignored compared to 5-20% DMSO. Based above reasons, we revised the manuscript as " The cell in the cell culture medium alone and supplemented with 5-20% dimethyl sulfoxide (DMSO) were usually chosen as negative and positive control groups, respectively " on page 21, lines 228-230.

Thank you again for your valuable comments on our manuscript, which have great benefits for us.

Reviewer 3 Report

In this article, the authors aim to summarize the cytotoxic effects of zinc and its alloys and try to identify the factors responsible for the toxicological responses. Zinc-based aggregates have been developed and investigated in clinical studies as potential implant materials due to their excellent biocompatibility and mechanical properties in vivo. However, a clear explanation of the cellular exclusivity of Zn and its alloy materials has been missing, so the authors collected 83 papers to investigate and summarize the relevant contents in this manuscript.

some minor questions need to be addressed or clarified:

(1) Can the authors describe the common criteria for standardized assessment of cytotoxicity of zinc and zinc alloys? In the manuscript, the authors show a large number of data collected and evaluated, but with relatively high variability.

(2) The authors point out that there are potential effects on the toxicity of zinc-based BMs due to variations in the mechanical properties of the metal, changes in the microstructure of the metal surface, and material processing techniques. Could they have been more precise in giving examples and some cases/examples from the reviewed articles.

(3) Three major categories of factors affecting the cytotoxicity of zinc and its alloys are summarized in the paper. Can these three be classified in the literature to derive adaptive rules that can be used as criteria for future assessment of the cytotoxicity of zinc and its alloys.

(4) At the end of the paper, could the authors give some outlook on the future development of zinc-based materials.

Author Response

Comments

In this article, the authors aim to summarize the cytotoxic effects of zinc and its alloys and try to identify the factors responsible for the toxicological responses. Zinc-based aggregates have been developed and investigated in clinical studies as potential implant materials due to their excellent biocompatibility and mechanical properties in vivo. However, a clear explanation of the cellular exclusivity of Zn and its alloy materials has been missing, so the authors collected 83 papers to investigate and summarize the relevant contents in this manuscript.

some minor questions need to be addressed or clarified:

Point 1: Can the authors describe the common criteria for standardized assessment of cytotoxicity of zinc and zinc alloys? In the manuscript, the authors show a large number of data collected and evaluated, but with relatively high variability.

Response 1: Thanks to the reviewer for raising this question. Till now, we consider the ISO 10993-12: 2021 and 10993-5: 2009 standards to be regarded as relatively thorough general standards for zinc-based metal cytotoxicity assessment, although it is not tailor-made for biodegradable metals. To develop a generic cytotoxicity assessment standard for zinc-based BMs is a large challenge that requires multiple experts and their teams to work together to overcome. Thus, we cannot describe the common criteria for cytotoxicity assessment in Zn-based BMs. In our view, the specificity of biodegradable metals needs to be fully taken into account, such as assessing not only the cytotoxicity of extracts but also the insoluble particulate degradation products.

There is relatively high variability among included studies. We think the reason for this lies more in the fact that plenty of researchers are not testing according to existing international standards or not reporting some key procedures, such as the extract ratio.

Point 2: The authors point out that there are potential effects on the toxicity of zinc-based BMs due to variations in the mechanical properties of the metal, changes in the microstructure of the metal surface, and material processing techniques. Could they have been more precise in giving examples and some cases/examples from the reviewed articles.

Response 2: We feel great thanks for your insightful suggestions on our manuscript. There are several examples mentioned in the manuscript to demonstrate our viewpoint about the different processing technologies changes in the microstructure of the metal surface, and might have an effect on cytotoxicity ultimately (lines 265-268, lines 269-271, and lines 273-275). And we try to explain these examples in detail and give specific data to prove our point of view.

  1. [52] Shi, Z. Z.; Gao, X. X.; Chen, H. T.; Liu, X. F.; Li, A.; Zhang, H. J.; Wang, L. N. Enhancement in mechanical and corrosion resistance properties of a biodegradable Zn-Fe alloy through second phase refinement. Sci. Eng. C Mater. Biol. Appl. 2020, 116, 111197. DOI:10.1016/j.msec.2020.111197.

The alloys produced through the conventional casting and BCWC were designated as Zn-0.3Fe (C) and Zn-0.3Fe (N), respectively.

This study demonstrated that more uniform corrosion and reduced corrosion rate were brought about by the refinement of the second phase of the bottom circulating water-cooled casting method, leading to higher cell viability than with conventional casting in 100% extracts.

  1. [20] Shen, C.; Liu, X.; Fan, B.; Lan, P.; Zhou, F.; Li, X.; Wang, H.; Xiao, X.; Li, L.; Zhao, S.; Guo, Z.; Pu, Z.; Zheng, Y. Mechanical properties, in vitro degradation behavior, hemocompatibility and cytotoxicity evaluation of Zn–1.2Mg alloy for biodegradable implants. RSC Adv. 2016, 6, 86410-86419. DOI:10.1039/c6ra14300h.

(a) as-cast and (b) as-extruded Zn–1.2Mg alloy.

This study showed the as-extruded Zn-1.2Mg alloy has lower cell survival than its as-cast alloys. The authors inferred the reason for lower cell survival was due to its higher corrosion rate resulting in a higher concentration of Zn ions (as shown in the table).

  1. [31] Tong, X.; Shi, Z.; Xu, L.; Lin, J.; Zhang, D.; Wang, K.; Li, Y.; Wen, C. Degradation behavior, cytotoxicity, hemolysis, and antibacterial properties of electro-deposited Zn-Cu metal foams as potential biodegradable bone implants. Acta Biomater. 2020, 102, 481-492. DOI:10.1016/j.actbio.2019.11.031.

This study showed the corrosion resistance of the electro-deposited Zn matrix could be dramatically improved by appropriate heat treatment, resulting in decreased released metal ions and a higher cytocompatibility.  

  1. [81] Chen, C.; Yue, R.; Zhang, J.; Huang, H.; Niu, J.; Yuan, G. Biodegradable Zn-1.5Cu-1.5Ag alloy with anti-aging ability and strain hardening behavior for cardiovascular stents. Mater. Sci. Eng. C Mater. Biol. Appl. 2020, 116, 111172. DOI:10.1016/j.msec.2020.111172.

This study showed that the appropriate plastic processing (as-extruded method) decreased released metal ions and therefore caused a higher cytocompatibility.

Based on the above examples, we explained these phenomena on page 22, lines 259-261, “Various processing techniques were used to improve the mechanical performance of Zn-based BMs. However, these techniques also changed the material microstructure, possibly influencing their corrosion behavior and biocompatibility”.

Point 3: Three major categories of factors affecting the cytotoxicity of zinc and its alloys are summarized in the paper. Can these three be classified in the literature to derive adaptive rules that can be used as criteria for future assessment of the cytotoxicity of zinc and its alloys.

Response 3: Thanks for your constructive comments. Regarding the factors relevant to the cytotoxicity assessment of zinc-based metals, we have divided them into three broad categories in general. We agree with the reviewer that it could be beneficial to derive adaptive rules if the three major categories of factors could be classified in the included literature. The cytotoxicity data available for zinc-based BMs are largely diverse. In fact, cytotoxicity assessment is an entirety, and each study involves these three categories of factors. So, it is difficult to classify these factors in the included literature. The question you raised is very enlightening, making us ponder if we could come up with some constructive ideas on cytotoxicity assessment criteria for zinc-based BMs based on this perspective in the subsequent studies. Therefore, we seek the reviewer’s understanding.

Point 4: At the end of the paper, could the authors give some outlook on the future development of zinc-based materials.

Response 4: Thanks for your great suggestion. In this work, we have only focused on cytotoxicity studies of biodegradable zinc-based metals. Therefore, we gave a shallow outlook on cytotoxicity studies development of Zn-based BMs on Page 21 lines 454-455, "A standardized in vitro toxicity assessment system for biodegradable metals is still lacking and further development is required." In addition, we also wish researchers will strictly follow the assessment criteria and report the test procedure in as much detail as possible to make the study data more informative in the future. We accepted your kind guidance and re-summarized the outlook for the development of Zn-based BMs at the end of the manuscript in the submitted revised version (Page 26, lines 455-459).
